# Uncovering Divergence in Gene Expression Regulation in the Adaptation of Yeast to Nitrogen Scarcity

Carlos A. Villarroel,[a,b,c] Macarena Bastías,[d] Paulo Canessa,[a,d] Francisco A. Cubillos[a,b]

aANID, Programa Iniciativa Científica Milenio, Instituto Milenio de Biología Integrativa (iBio), Santiago, Chile
bDepartamento de Biología, Facultad de Química y Biología, Universidad de Santiago de Chile, Santiago, Chile
cLaboratorio Interacciones Insecto-Planta, Instituto de Ciencias Biológicas, Universidad de Talca, Talca, Chile
dCentro de Biotecnología Vegetal, Facultad de Ciencias de la Vida, Universidad Andres Bello, Santiago, Chile

**ABSTRACT** *Saccharomyces cerevisiae* rewires its transcriptional output to survive stressful environments, such as nitrogen scarcity under fermentative conditions. Although divergence in nitrogen metabolism among natural yeast populations has been reported, the impact of regulatory genetic variants modulating gene expression and nitrogen consumption remains to be investigated. Here, we employed an F1 hybrid from two contrasting *S. cerevisiae* strains, providing a controlled genetic environment to map *cis* factors involved in the divergence of gene expression regulation in response to nitrogen scarcity. We used a dual approach to obtain genome-wide allele-specific profiles of chromatin accessibility, transcription factor binding, and gene expression through ATAC-seq (assay for transposase accessible chromatin) and RNA-seq (transcriptome sequencing). We observed large variability in allele-specific expression and accessibility between the two genetic backgrounds, with a third of these differences specific to a deficient nitrogen environment. Furthermore, we discovered events of allelic bias in gene expression correlating with allelic bias in transcription factor binding solely under nitrogen scarcity, where the majority of these transcription factors orchestrates the nitrogen catabolite repression regulatory pathway and demonstrates a *cis* × environment-specific response. Our approach allowed us to find *cis* variants modulating gene expression, chromatin accessibility, and allelic differences in transcription factor binding in response to low nitrogen culture conditions.

**IMPORTANCE** Historically, coding variants were prioritized when searching for causal mechanisms driving adaptation of natural populations to stressful environments. However, the recent focus on noncoding variants demonstrated their ubiquitous role in adaptation. Here, we performed genome-wide regulatory variation profiles between two divergent yeast strains when facing nitrogen nutritional stress. The open chromatin availability of several regulatory regions changes in response to nitrogen scarcity. Importantly, we describe regulatory events that deviate between strains. Our results demonstrate a widespread variation in gene expression regulation between naturally occurring populations in response to stressful environments.

**KEYWORDS** yeast, allele-specific, ATAC-seq, nitrogen, regulatory divergence, wine, fermentation, gene regulation, genetics, natural variation

Uncovering the molecular configurations that underlie gene expression divergence in adaptation to stressful environments constitutes a relevant genetic quest. The yeast *Saccharomyces cerevisiae* provides an excellent genetic model to investigate the link between the regulatory divergence of sequences and environmental fluctuations (1). Yeast cells undergo extensive reprogramming of their gene expression profiles to withstand different environmental stresses; of these, the transcriptional response to nitrogen scarcity has been comprehensively described (2–5). Yeast fitness strongly

Address correspondence to Francisco A. Cubillos, francisco.cubillos.r@usach.cl.

mSystems®

depends on the availability of preferred nitrogen sources, and changes in such nutritional signals trigger the immediate transcriptional rewiring of nitrogen metabolism. In yeast, at least four pathways are involved in regulating nitrogen metabolism, with the nitrogen catabolite repression (NCR) pathway (6) being the main orchestrator of the response to nitrogen starvation. Importantly, extensive natural variation in nitrogen consumption (7, 8) and starvation tolerance under wine fermentative conditions (3) have been found among yeast populations, demonstrating significant divergence in the regulatory mechanisms involved in the NCR pathway (9).

Yeast strains have different nitrogen consumption profiles, amino acid preferences, and tolerance to nitrogen scarcity during wine fermentation (7, 10). In general, wine strains have rapid and efficient nitrogen consumption profiles compared to wild strains (7, 10–12). In this way, winemaking strains exhibit physiological adjustments to poor nitrogen environments while displaying good fermentation performance, a feature that is a hallmark of domestication (10, 13–15). Using quantitative trait locus (QTL) approaches, several genes involved in differences in nitrogen consumption have been mapped (3, 7, 9, 16), including important nodes of the NCR pathway such as *GTR1*, which encodes a subunit of a TORC1-stimulating GTPase (12) and *RIM15*, involved in cell proliferation in response to nutrients (17). Nevertheless, the regulation of gene expression and the modifying role of polymorphic transcription factors (TFs) in response to nitrogen scarcity in wine strains remain to be elucidated.

Transcriptional divergence originates from genetic variants, which can be identified through mapping of expression QTLs (eQTLs) (1, 18). eQTLs might regulate the adjacent allele (*cis* eQTL) or affect one or multiple distant genes (*trans* eQTL). First-generation (F1) hybrids constructed from individuals of divergent lineages offer a refined approach to map *cis* factors responsible for expression divergence (19–22). In this F1 hybrid setup, the *trans* component is neglected, as *trans* eQTLs affect both parental alleles in the same way, therefore cancelling potentially different contributions. On the other hand, *cis* effects will remain allele specific (23). Moreover, differences in the expression of each allele (allele-specific expression [ASE]) are explained by the allele's local variants, which might control the physical accessibility of its promoter or regulatory region (24). This could be achieved by modulating the affinity of TF binding sites or affecting the regulation of the encoded RNA at a posttranscriptional level (25–28). Numerous studies have extensively quantified ASE in different model organisms (29–33). However, most of these studies have not incorporated a genome-wide experimental approach that assesses the *cis*-regulatory mechanisms underlying allelic expression variation. Coupling massive mRNA sequencing with assays that cut DNA *in vivo* at physically accessible chromatin regions, such as ATAC-seq (assay for transposase accessible chromatin), can portray a whole-genome profile of DNA accessibility to transcriptional regulators (34). In addition, ATAC-seq can also provide a genome-wide survey of transcription factor binding (TFB), allowing the *in silico* footprinting of TFB at open chromatin regions (35). ATAC-seq has been employed in yeast to investigate regulatory mechanisms driving aging (36), metabolism and cell division (37), pathogenesis (38), and cold adaptation in an interspecies hybrid (21). Recent studies in mouse crosses have incorporated assays that profile allelic differences at the transcriptional regulatory level (39–41), demonstrating the suitability of these techniques to measure allelic imbalance in the regulation of gene expression.

Herein, we measure allelic imbalance occurring at the level of gene expression, chromatin accessibility, and TFB in an F1 hybrid between two divergent *S. cerevisiae* strains. We evaluated whether differences in nitrogen consumption between a winemaking strain (DBVPG6765), and an undomesticated strain (YPS128) isolated from an oak tree were due to *cis*-regulatory variants modulated through environments differing in nitrogen availability. We report numerous events of allelic differences in chromatin accessibility between these two strains, remarkably few of which directly correlate with ASE. Furthermore, we show that one third of the allelic differences in gene expression and accessibility occur only under low nitrogen. By performing allele-specific TFB

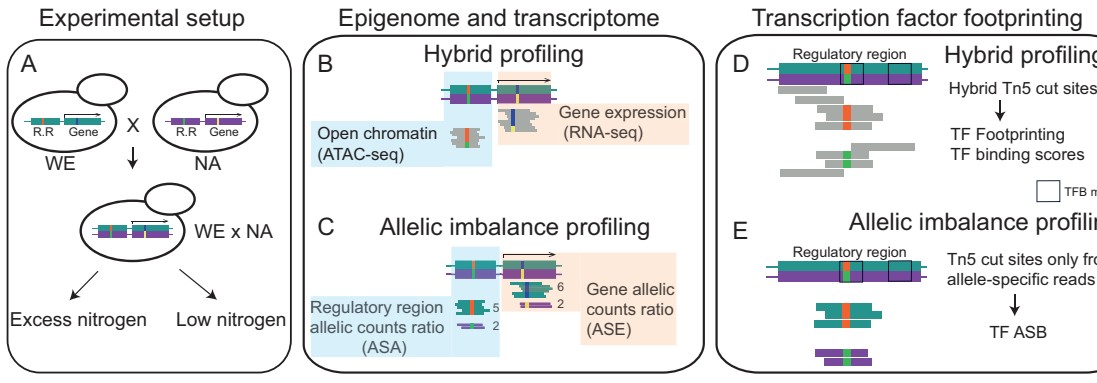

**FIG 1** Allele-specific expression and chromatin accessibility profiling to reveal molecular mechanisms orchestrating gene expression divergence in response to nitrogen scarcity. (A) Representative haploid strains of the WE (Wine European) and NA (North American) lineages of *S. cerevisiae* were selected to construct a WE × NA hybrid which was used to perform fermentations in synthetic wine media under low and excess nitrogen conditions (SM60 and SM300, respectively). Large boxes denote a *cis*-regulatory region (R.R) and an open reading frame (ORF) (arrow on top), while smaller boxes depict polymorphisms occurring in each genetic background. (B) RNA-seq and ATAC-seq were employed to profile the hybrid's transcriptome and open chromatin landscape at regulatory regions between nitrogen conditions. Gray bars denote the common pool of hybrid reads. (C) RNA-seq and ATACseq reads that mapped to parent-specific single-nucleotide polymorphisms (SNPs) were used to determine allele-specific expression (ASE) and allele-specific accessibility (ASE and ASA, respectively). Reads highlighted in colors indicate different pools according to their parental origin. (D) ATAC-seq cut sites were used to infer *in silico* footprinting of transcription factor binding (TFB) in the hybrid. (E) Allele-specific ATAC-seq alignments were separated by parental origin to obtain allele-specific ATAC-seq cut sites which were used to infer allele-specific binding (ASB) from TFB *in silico* footprinting.

footprinting, we reveal TFs that potentially drive allelic expression differences, some of which have not been previously related to the regulation of nitrogen metabolism.

## RESULTS

**A differential response to nitrogen scarcity is observed in two divergent yeast strains.** To identify *cis*-regulatory variants driving gene expression divergence in the adaptation of yeast to nitrogen scarcity, we performed allele-specific expression and accessibility (ASE and ASA, respectively) employing RNA-seq (transcriptome sequencing) and ATAC-seq (assay for transposase accessible chromatin) in a *S. cerevisiae* cross (Fig. 1). To construct the F1 hybrid, we selected the DBVPG6765 and YPS128 strains, hereinafter referred to as WE (Wine European) and NA (North American), respectively. We chose these two genetic backgrounds due to their extreme nitrogen consumption profiles reported in former studies (7, 10–12) and the genetic distance between both strains (on average 1 single-nucleotide polymorphism [SNP] every 148 bp, 0.6% sequence divergence) allowing us to properly perform allele-specific analyses. Our previous results demonstrated that the WE strain consumed larger amounts of total yeast assimilable nitrogen (YAN) under wine fermentation conditions, while NA exhibited lower yields. To evaluate the response of these two genetic backgrounds to low nitrogen, the WE × NA F1 hybrid was grown under low or excess nitrogen concentrations (SM60 and SM300, respectively; see Materials and Methods).

First, we assessed nitrogen consumption kinetics in the WE × NA hybrid and the parental strains under both fermentation conditions. We sampled fermentations at an early time point, i.e., 14 h after synthetic grape must inoculation. Under excess nitrogen conditions, the parental NA strain exhibited lower consumption levels of total YAN than the hybrid and WE strains (*P* value < 1e−04, analysis of variance [ANOVA]) (Fig. 2A). Although we did not find higher total YAN consumption in the hybrid, after examining each nitrogen source, we found that the hybrid strain consumed higher levels of serine and alanine in excess nitrogen compared to both parental strains (*P* value < 0.05, ANOVA) (Fig. 2B). We formally estimated heterosis and found that the consumption of six amino acids had a heterosis coefficient higher than 1 (Fig. 2C). Among those amino acids that are rapidly consumed in the hybrid, we found all amino acids transported by Agp1p/Gnp1p (serine, threonine, and glutamine) (Fig. 2C). In contrast to excess nitrogen, under low nitrogen conditions, YAN was almost depleted from

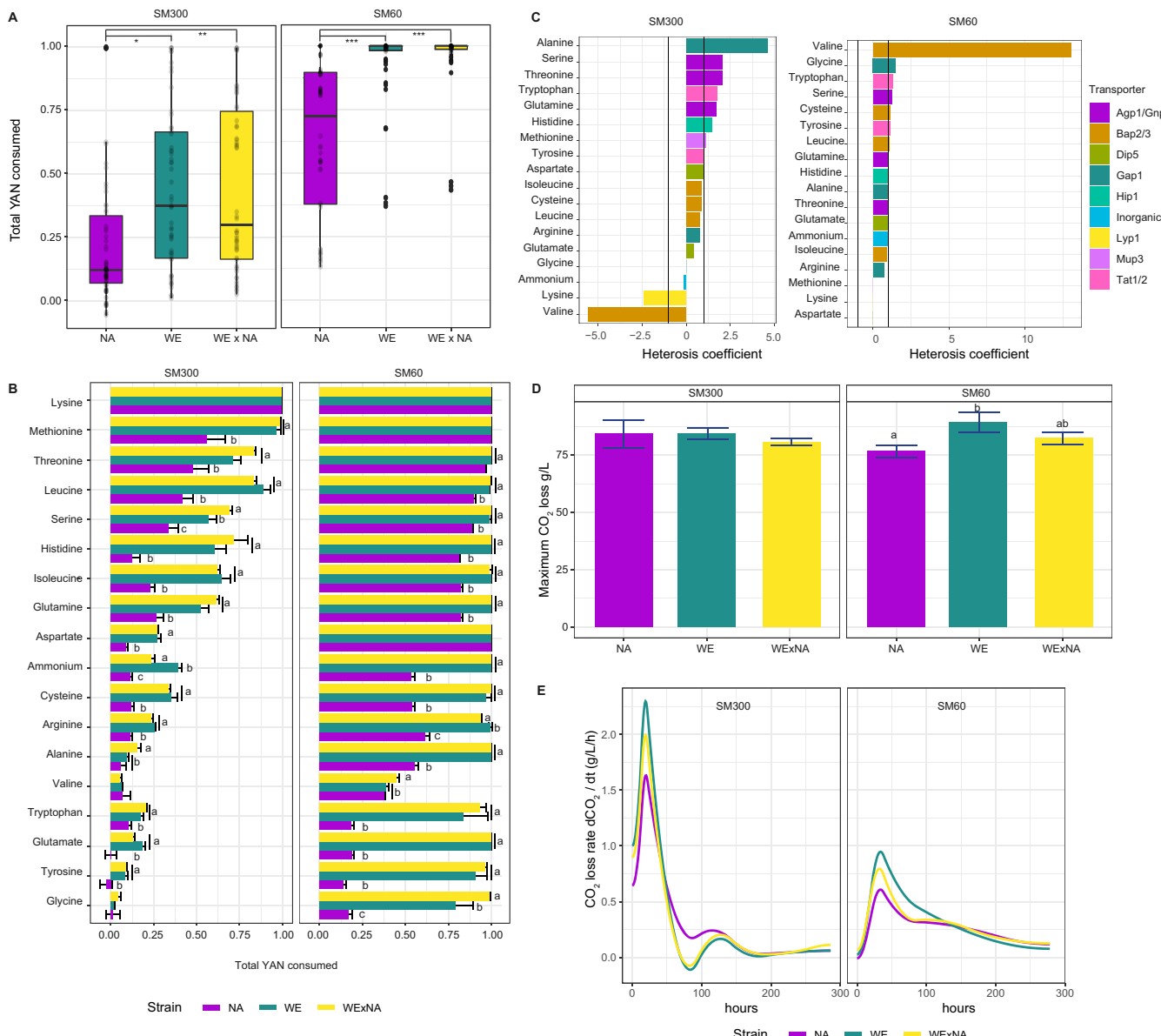

**FIG 2** Fermentation kinetics and nitrogen consumption in WE and NA parental strains and the WE × NA hybrid. (A) The boxplots show the total yeast assimilable nitrogen (YAN) consumed by the NA, WE, and WE × NA strains after 14 h of fermentation in synthetic wine must containing excess or low nitrogen (SM300 and SM60, respectively). Consumption of each amino acid is shown as dots. Asterisks denote significance difference (Tukey *post hoc*) as follows: *. $P < 0.05$; **. $P < 0.001$; *** $< 1e-5$. (B) The bars indicate consumption fraction of each amino acid in SM300 and SM60 after 14 h of fermentation by the WE × NA (yellow), WE (green), and NA (purple) strains. Letters indicate significant differences between groups (ANOVA, Tukey *post hoc* test). Error bars show the standard deviations (SD). (C) The bars show the heterosis coefficient for the consumption of each amino acid in SM300 and SM60. Values > 1 indicate best parent heterosis; values < −1 indicate worst-parent heterosis. (D) Total production of $CO_2$ at the end of fermentation process of each indicated strain and culture condition. Letters indicate significant differences among groups (ANOVA, Tukey *post hoc* test) and error bars denote SD. (E) Fermentation rates of the WE, NA, and WE × NA strains in SM300 and SM60 culture conditions.

the media after 14 h of fermentation, particularly by the WE and WE × NA strains, with the NA strain exhibiting lower total YAN consumption (*P* value < 0.05, ANOVA, Fig. 2A). In addition, we found differences in the consumption kinetics for 14 amino acids when comparing the hybrid and the two parental strains (*P* value < 0.05, ANOVA), together with heterosis in valine consumption (Fig. 2B and C). After 14 days of fermentation, the three genetic backgrounds showed no differences in their fermentation performance under excess nitrogen (Fig. 2D), though under low nitrogen conditions, the NA strain had the lowest total $CO_2$ loss (*P* value WE − NA = 0.01; *P* value WE × NA − NA = 0.09, ANOVA), which indicates a stronger negative effect of nitrogen scarcity on the fermentation performance of the NA strain and nitrogen starvation stress (10, 42).

Furthermore, low nitrogen affected all strains' fermentation kinetics compared to excess nitrogen culture conditions, diminishing the maximum fermentation speed by 63%, 58%, and 60% in NA, WE, and WE × NA strains, respectively (Fig. 2E). These results demonstrate that the genotype affects fermentation performance during nitrogen scarcity and suggest a dominant inheritance of efficient nitrogen consumption in yeast.

**Low correlation between gene expression and chromatin accessibility in response to fermentation under low nitrogen.** We collected mRNA from the WE × NA hybrid after 14 h of fermentation under low or excess nitrogen culture conditions. We chose this time point due to the primary consumption of preferred nitrogen sources under excess nitrogen (NCR suppressed state) and the complete YAN consumption under low nitrogen, likely triggering a nitrogen starvation stress response (NCR active state). Hence, significant differences in gene expression and regulation between environments and genetic backgrounds were expected (Fig. 2B) (10, 42). We found a total of 3,719 DEGs (differentially expressed genes) between conditions, of which 1,842 and 1,877 were upregulated or downregulated, respectively, in response to low nitrogen (false discovery rate [FDR] of <0.05 [see Table S1a in the supplemental material]). Among the upregulated genes, we found 69 DEGs previously classified as NCR sensitive (43). Interestingly, 21 of these NCR-sensitive genes were determined among the top 30 genes that were most induced by low nitrogen (Table S1b). Enriched biological processes among low nitrogen induced genes were related to transport, energy generation, detoxification, and oxido-reduction (Table S2a), while genes associated with ribosomal biogenesis were significantly enriched among downregulated genes (Table S2b).

To profile the chromatin accessibility landscape in response to low nitrogen, we performed ATAC-seq in hybrid replicates collected in the aforementioned culture conditions (for ATAC-seq quality control [QC] plots, see Fig. S1 in the supplemental material). First, we assessed the correlation between gene expression and chromatin accessibility. We found a significant, though moderate correlation (Spearman $R = 0.28$ and $R = 0.30$ for excess and low nitrogen, respectively; $P$ value $< 1e-99$), between the amount of gene expression (reads per kilobase per million [RPKM]) and the corresponding ATAC-seq signal up to 1,000 bp upstream of the transcript start site (TSS). A sliding window analysis showed that the highest and most significant correlation between gene expression of a given gene and its nearest ATAC-seq signal differed between nitrogen levels (Fig. 3A). In particular, for excess nitrogen, we found a narrower signal region upstream of the TSS ($-350$ to $-50$ bp, Spearman $R = 0.32$, $P$ value $< 1e-135$), while we estimated a wider region for low nitrogen ($-400$ to $+50$ bp, Spearman $R = 0.33$, $P$ value $< 1e-140$) (white squares in Fig. 3A). These results show that different segments within the gene's regulatory regions respond to changing nitrogen concentrations. For the analyses shown hereafter, to evaluate the corresponding ATAC-seq signal, we considered a regulatory region of 400 bp upstream of the TSS of each gene. We based the selection of the regulatory region size on two reasons. (i) It correlates well with gene expression under low and excess nitrogen conditions. (ii) Another study previously used a similar region size (44).

By analyzing 5,625 regulatory regions, we found an increase in chromatin accessibility in 376 differentially accessible regions (DARs) and a decrease in 875 DARs (FDR < 0.05) under low nitrogen conditions, representing 22.2% of the analyzed regions (Fig. 3B and Table S1). Generally, chromatin accessibility fold changes between conditions were lower in absolute magnitude for DARs (mean $| \log_2$ fold change [FC] $| = 0.38$) than those expression fold changes observed for DEGs (mean $| \log_2$ FC $| = 0.95$) ($P$ value $= 0.001$, Mann-Whitney-Wilcoxon test, Fig. 3C). We found that regions regulating genes involved in glucose transport and urea metabolism were less accessible under low nitrogen conditions (Table S2c), while regions regulating genes involved in the metabolism of nonpreferred nitrogen sources were more accessible when subjected to nitrogen scarcity (Table S2d).

We examined whether differential gene expression between conditions correlated with chromatin accessibility. We found that 444 DEGs and DARs showed downregulation and lower accessibility, and conversely, an induction in transcript levels with higher chromatin accessibility. Downregulated genes within this positively correlated set were enriched in processes related with cytoplasmic translation and vitamin biosynthesis,

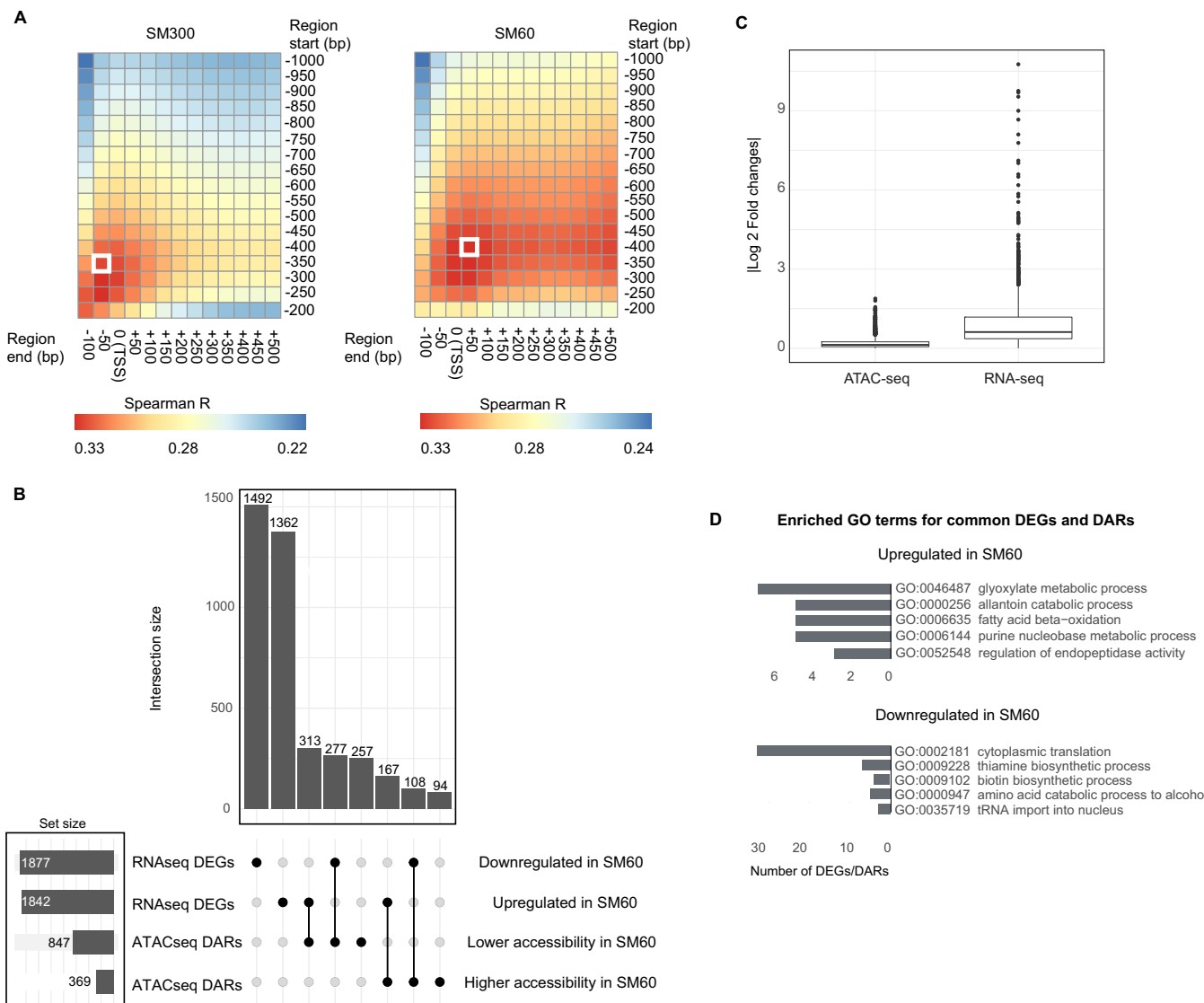

**FIG 3** Hybrid transcriptome and regulatory region accessibility in response to fermentation in low nitrogen wine must. (A) The heatmaps show the Pearson correlation between ATAC-seq coverage and total gene expression of the WE × NA hybrid after 14 h of fermentation in SM300 and SM60. For each gene, different regulatory region segments were evaluated using a sliding window approach starting from 1,000 bp upstream of the transcription start site (TSS) to 200 bp upstream of the TSS, and ending from 100 bp upstream of the TSS to 500 bp downstream of the TSS. The white boxes indicate the regions with the highest correlation in SM300 and SM60. (B) Upset plot showing the number of differentially expressed genes (DEGs) and differentially accessible regions (DARs), the set sizes, and the intersect among combinations. (C) Average absolute log₂ fold change of the ATAC-seq and RNA-seq data sets. (D) Number of genes occurring in gene ontology (GO) terms (biological processes) enriched among common DEGs and DARs.

while upregulated genes were enriched for catabolism of allantoin and glyoxylate (Fig. 3D). In contrast, a large number of DEGs related with ribosome biogenesis showed no differences in accessibility at their regulatory regions across nitrogen conditions, despite being concertedly less expressed in response to low nitrogen (Fig. S2A). In addition, among non-DEGs, we found differential accessibility in genes related with sugar transport (Fig. S2B). These results indicate that approximately 12% of gene expression differences in the yeast genome positively correlated with chromatin accessibility differences between conditions. We would like to highlight that RNA-seq and ATAC-seq experiments were performed independently, although carefully sampling after the same period of time. Still the absence of correlation between gene expression fold changes and chromatin accessibility has been previously observed in yeast (21, 45). In summary, chromatin accessibility results complement those obtained by RNA-seq by providing novel evidence of regulatory rewiring under contrasting nitrogen conditions.

**Allele-specific expression and chromatin accessibility under excess and low nitrogen conditions.** We compared the transcriptional response to low nitrogen of each parental genome within a shared (WE × NA) *trans* environment. We found that parental genomes display a highly similar transcriptional response to low nitrogen (Pearson $R = 0.911$, $P$ value $< 2.2e-16$), although differences in ASE could still be found. Therefore, we investigated ASE by performing a binomial test of read counts at 21,647 SNPs in 3,923 genes (1,732 genes did not have SNPs or only had SNPs with fewer than 10 counts). Differential ASE was determined in 543 genes, of which 147 and 245 were solely found in excess nitrogen and low nitrogen, respectively, while 153 were significant under both conditions (Fig. 4A and B and Table S3), the majority of which (152) maintained the ASE direction, i.e., bias in expression favored the same genotype in both conditions. Allelic ratios in both conditions were correlated (Pearson $R = 0.69$, $P$ value $< 1e-15$), although they were lower than those described in other ASE assays in yeast (22, 46). Accordingly, a significant portion of ASE differences in yeast under contrasting nitrogen conditions are dependent on the *cis*-genotype × environment interaction, rather than primordially on the *cis*-genotype configuration.

Further dissection of the ASE of strongly upregulated genes under low nitrogen conditions (FC $> 3$, 251 DEGs) allowed us to identify key variants involved in NCR. For example, we found high WE allelic expression compared to the NA variant for the amino acid permeases *GAP1* and *PUT4* and for the ammonium permease *MEP2* (Fig. 4B). In contrast, the NA allele encoding the NCR-sensitive allantoate transporter (*DAL5*) showed a strong bias in expression compared to the WE allele. These results indicate that genetic variants in ammonium and specific amino acid permeases are overexpressed in wine strains when facing nitrogen scarcity, putatively due to *cis*-regulatory variants.

To assess these differences, allele-specific accessibility was determined in each parental genome in response to low nitrogen within the shared *trans* environment (Fig. 1C). Interestingly, we found a moderate correlation between the chromatin accessibility response of both parental genomes (Pearson $R = 0.529$, $P$ value $< 1e-15$), which was lower than that found for the gene expression response, suggesting a greater impact of environmentally dependent differences in *cis* regulation acting on chromatin accessibility than on gene expression. We tested allelic imbalance in chromatin accessibility (binomial test) and were able to quantify ASA in 15,333 SNPs belonging to 4,822 regulatory regions. A total of 252 regions exhibited differential ASA, and from this, 113 and 69 regulatory regions showed ASA under either low or excess nitrogen, respectively (Table S3). ASA was moderately correlated between nitrogen conditions (Pearson $R = 0.61$, $P$ value $< 1e-15$), similar to what was found for ASE. Importantly, all regions that showed ASA in both conditions (70 in total) maintained the imbalance direction, i.e., bias in accessibility favored the same genotype in both conditions (Fig. 4C). Among enriched biological processes in regions that showed ASA in low nitrogen, we found more accessible regulatory regions associated with the expression of aldehyde metabolic process genes and the response to oxidative stress in the WE and NA genomes, respectively (Table S4c and S4d).

Taking the differential ASE and ASA data sets together, we assessed the intersection between them under each condition. Under excess nitrogen, 13 genes displaying ASE also showed ASA in the same direction, while 5 genes showed ASA in the opposite direction (Fig. 4D). Moreover, under low nitrogen, 28 genes showed ASE and ASA in the same direction, while 10 displayed ASA and ASE in opposite directions (Fig. 4E). Overall, these results indicate a convergence between ASA and ASE in 41 genes, where open chromatin alleles had greater expression levels. However, only 5% and 9% of genes in ASE coincided with allelic imbalance in accessibility occurring in the same direction in excess or low nitrogen, respectively, while the large majority of genes exhibiting ASE did not show ASA levels and vice versa. These results suggest that additional regulatory mechanisms, such as differences in TFB, could influence differences in allelic expression.

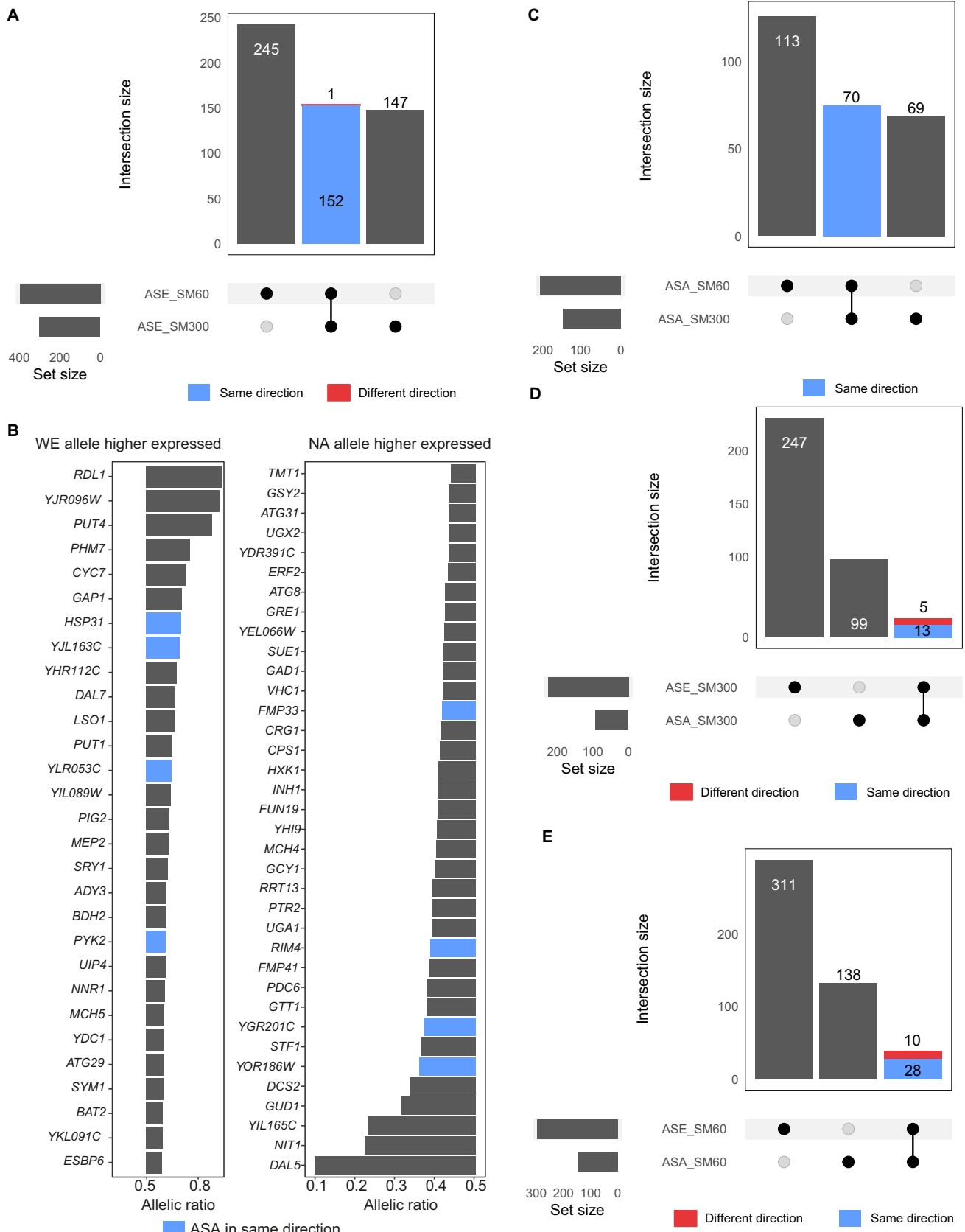

**FIG 4** Allelic imbalance in accessibility and gene expression in the WE × NA hybrid. (A) Upset plot showing the number of genes in differential allelic imbalance under SM300 and SM60 in the WE × NA hybrid and the intersection between conditions. Blue bars indicate that allelic ratios were

**Allelic imbalance in transcription factor binding reveal environment-dependent regulatory mechanisms driving gene expression divergence.** To explore the link between TFB and allele-specific expression differences occurring in the WE × NA hybrid, we used ATAC-seq to infer TFB and generate genome-wide binding scores for 141 TFs. First, we inspected fold changes of TFB scores in the hybrid across conditions, discovering a strong correlation with ATAC-seq fold changes (Pearson $R = 0.71$, $P$ value $< 1e-130$) (Fig. 5A). However, no overall correlation between fold changes of TFB scores and gene expression differences between conditions was found (Pearson $R = 0.072$, $P$ value $1.58e-31$) (Fig. 5A), suggesting a complex interplay between global TFB and gene expression, while also evidencing a direct effect of chromatin accessibility on TFB scores. We reexamined the correlation between TFB scores and gene expression, but this time individually for each TF. In this way, we found a significant correlation for 11 out of 141 TFs (Pearson $R > 0.2$, FDR $< 0.1$) (Table S5a). The Cst6p TF showed the highest correlation between TFB and gene expression (Pearson $R = 0.36$, FDR $= 0.03$) (Fig. 5B). Cst6p encodes a basic leucine zipper TF involved in the stress response (47), and our results suggest a role for Cst6p in response to low nitrogen conditions. Moreover, for all TFs tested, the binding scores were highly correlated with their ATAC-seq fold changes, ranging from Dal80p (Pearson $R = 0.882$, FDR $= 3.08$ e-25) to Fkh2p (Pearson $R = 0.50$, FDR$= 3.14$ e$-6$) (Table S5a).

Next, we inferred allele-specific binding (ASB) by estimating the TFB scores for each parental strain (Fig. 1E). We found a significant correlation between ASB and ASA (for low nitrogen; Pearson $R = 0.53$, $P$ value $< 1e-130$). However, no genome-wide correlation was found between ASB and ASE under any condition (for low nitrogen; Pearson $R = 0.09$, $P$ value $= 4.36e-47$). As previously done with the hybrid TFB scores, we also decided to inspect the correlation between ASB and ASE for each TF individually (Table S5b). Under low nitrogen, we observed that the allele-specific binding for six TFs correlated with ASE of their target genes (Pearson $R > 0.2$, FDR $< 0.05$). Among them, we found three out of the four GATA-type zinc finger TFs that participate in NCR regulation; i.e., Gat1p, Dal80p, and Gzf3p (Fig. 5C and Table S5b). Together with Gln3p, these GATA TFs share very similar binding motifs (shown in Fig. 5C), which suggest that these correlations might not be specific for any of these TFs in particular. In the case of Gln3p, a shorter and less informative GATA motif was used, which substantially increased the number of predicted binding sites compared to those of the other GATA factors, suggesting a stronger influence of false-positive binding sites on the lack of correlation between Gln3p ASB and the ASE of its target genes under low nitrogen (Pearson $R = 0.09$). We cross-referenced our Dal80p *in silico* binding data under nitrogen scarcity with a Dal80p chromatin immunoprecipitation-DNA sequencing (Chip-seq) data set performed under a similar stress condition (43), and found a strong agreement for Dal80p binding at 36 (53%) of 67 predicted bound promoters.

In addition to GATA factors, we found that Skn7p, Swi4p, Tos8p, Yap5p, Tod6p, and Yox1p TFs had a significant correlation between ASB and ASE under low nitrogen (Table S5b). When nitrogen is not limited, we found a significant correlation between ASE and ASB solely for Yap5p and Swi4p (Pearson $R > 0.2$, FDR $< 0.1$), and specifically for this condition for Yap7p (Table S5b). These results suggest a role for TFs that have not previously been associated with nitrogen metabolism in the adaptation of winemaking strains to nitrogen scarcity in grape must. Nevertheless, we expect that the significant correlations found between ASB and ASE were confounded by the allele-specific ATAC-seq signal, which prompted us to evaluate our ASB data in more detail.

**FIG 4** Legend (Continued)
maintained between conditions, while red bars indicate that allelic ratios were inverse between conditions. (B) The bar plots show the expression ratio of genes with significant ASE and induced by SM60 (FDR $< 0.05$, $\log_2$ fold change $> 3$). Genes with allelic ratios higher than 0.5 had higher expression of the WE allele, while genes with allelic ratios lower than 0.5 had higher expression of the NA allele. Blue bars indicate genes that also showed differential ASA in the same direction of ASE, i.e., bias favoring the same genotype in accessibility and expression. (C) Upset plot showing the numbers of regulatory regions in differential ASA in SM300 and SM60 and the intersection across conditions. (D and E) The upset plot shows the number of differential ASE and ASA in SM300 (D) and SM60 (E), and the intersection of these within the same condition.

mSystems®

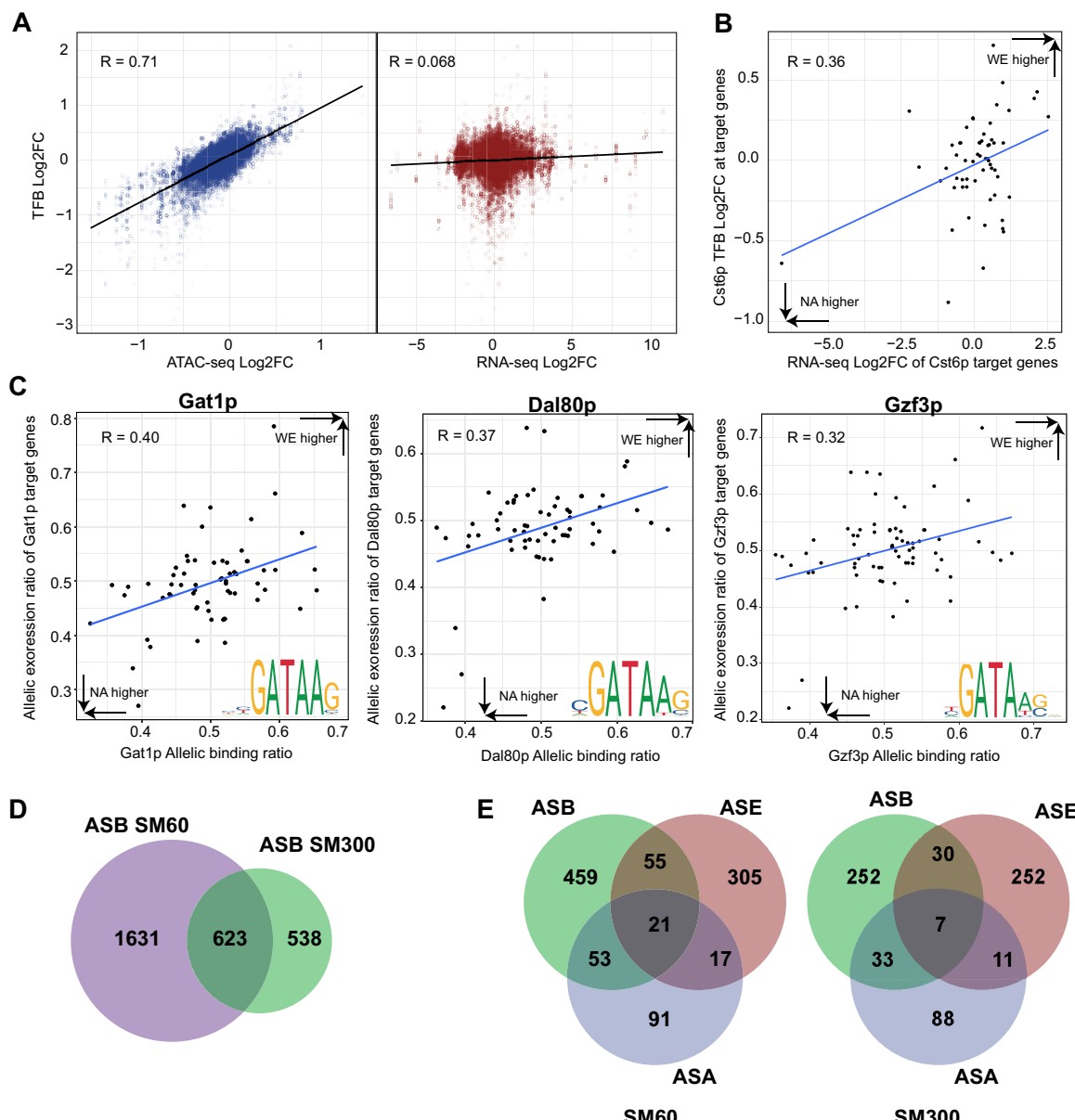

**FIG 5** Allele-specific differences in transcription factor binding. (A) Correlation between ATAC-seq log$_2$ fold change (Log2FC) and transcription factor binding (TFB) Log2FC (blue dots), and RNA-seq Log2FC and TFB Log2FC (red dots) in response to low nitrogen in the WE × NA hybrid. (B) Correlation between Cst6p binding scores and expression in response to low nitrogen of its target genes. (C) Correlation between the allelic expression ratio of the target genes of Gat1p, Dal80p, and Gzf3p and the allelic binding ratio of these transcription factors (TFs) at their target genes regulatory regions. Allelic ratios higher than 0.5 indicate higher expression or binding of the WE allele. Sequence logos indicate the motif used for each TF to scan binding sites throught the genome. (D) The Venn diagram displays the number of binding sites showing significant ASB in SM300 or SM60 and in both conditions. (E) The Venn diagrams indicate the number of regulatory regions having at least one binding site showing ASB and the number that were associated with regions showing ASE or ASA or both in SM60 and SM300.

We tested for significant differences in ASB (see Materials and Methods) and found that 623 out of 27,370 binding sites were differentially bound in both conditions (Fig. 5D and Table S5c and S5d). Furthermore, 513 (22%) and 293 (25%) of the binding sites displaying ASB overlapped with a SNP under low or excess nitrogen, respectively. This represents a higher proportion than that observed for all tested binding sites (16%), suggesting that binding sites cooccurring with SNPs were more likely to be differentially bound (chi-square test, *P* value = 7.3e−18).

Next, we wanted to identify TFs that are differentially bound to the WE or NA allele that might drive ASE in the absence of chromatin accessibility differences. Hence, we

focused our ASB analysis on those regulatory regions in which we did not find ASA, but their genes showed ASE. Under low nitrogen conditions, of the 311 genes that showed ASE (but not ASA), we found 55 regulatory regions displaying ASB, representing 17% of the ASE differences (Fig. 5E, FDR < 0.1). Also, 247 genes showed ASE but not ASA in excess nitrogen; of those, 30 showed ASB at their regulatory regions (Fig. 5E, FDR < 0.1). Moreover, many of the binding sites showing ASB did not alter allelic expression (Fig. 5E). These results demonstrate the additional contribution of ASB in the absence of ASA toward differences in allelic expression under low nitrogen conditions.

**Identification of *cis*-acting variants driving allelic expression differences in response to nitrogen scarcity.** We inspected in detail those DEGs whose expression was upregulated or downregulated by low nitrogen in the hybrid context, which also exhibited ASB and ASE but not ASA. We observed ASB in 23 and 13 regulatory regions, respectively, in DEGs induced or repressed under low nitrogen (Fig. 6A and Fig. S3, induced and repressed in low nitrogen, respectively). As an example of a binding site cooccurring with a SNP and displaying ASB, we show the TFB site for the transcriptional repressor Mot2p localized at the regulatory region of the *HXK1* gene, which encodes a hexokinase that phosphorylates hexoses for subsequent glycolysis (Fig. 6B). We found a 1.2-fold-higher binding for the WE allele than for the NA allele (FDR 0.08), consistent with higher expression of the *HXK1*-NA allelic variant under low nitrogen. The causal variant could be a SNP (A > G) affecting the binding motif of Mot2p in the NA genome. Another example is shown for a differentially bound region in the promoter of *STF1*, which codes for an accessory protein involved in the inhibition of the $F_1F_0$-ATP synthase complex, at which the binding sites of Gln3p and Pho2p colocalize with a SNP that might drive allelic differences in expression found for the *STF1* gene (Fig. 6C). Among downregulated DEGs, we observed differential binding at the motifs for the TFs Msn2p, Msn4p, and Rgm1p present in the promoter of *THI4*, which is involved in thiamine biosynthesis (Fig. S3B).

As an example of a binding site that is relatively far from a putative causal SNP but that exhibits large allelic binding differences, we found in the *MEP2* regulatory region (which encodes a ammonium permease) two sequentially occurring GATA-like sites for Gln3p, both of which showed higher binding scores for the WE allele than for the NA variant (1.93-fold higher in WE under low nitrogen, FDR < 0.003, Fig. 6D), coincident with the higher expression of the *MEP2*-WE allele under the same culture condition. Importantly, *MEP2* was highly induced under low nitrogen, and the allelic differences in binding and expression were significant only under limited nitrogen.

In summary, our results demonstrate that the specific identification of allele-specific TFB, together with differences in chromatin accessibility, can shed light onto novel molecular targets and mechanisms driving phenotypic differences between yeast strains.

## DISCUSSION

In this work, we describe genome-wide allelic imbalance events at three levels—gene expression, chromatin accessibility, and transcription factor binding—which allowed us to expose *cis* mechanisms driving the adaptation to low nitrogen fermentation in winemaking yeast. The transcriptional response of the WE × NA hybrid to low nitrogen resembled that of other studies of yeast under nitrogen stress (10, 43, 48, 49). However, the numerous differences found in allele-specific expression (ASE) and accessibility (ASA) suggest widespread variation in *cis* mechanisms regulating nitrogen metabolism among natural yeast populations. We found large differences in the gene expression and chromatin accessibility profiles in response to nitrogen scarcity, even though both strategies provided similarly enriched functional annotations among the concertedly regulated genes. We hypothesize that differences in accessibility translate into differences in gene expression at distinct times depending on the pathway involved. For example, we found increased accessibility and expression for genes associated with metabolism of nonpreferred nitrogen sources (e.g., allantoin and carnitine). In contrast, we found less accessibility in six genes encoding hexose transporters without the involvement of differential

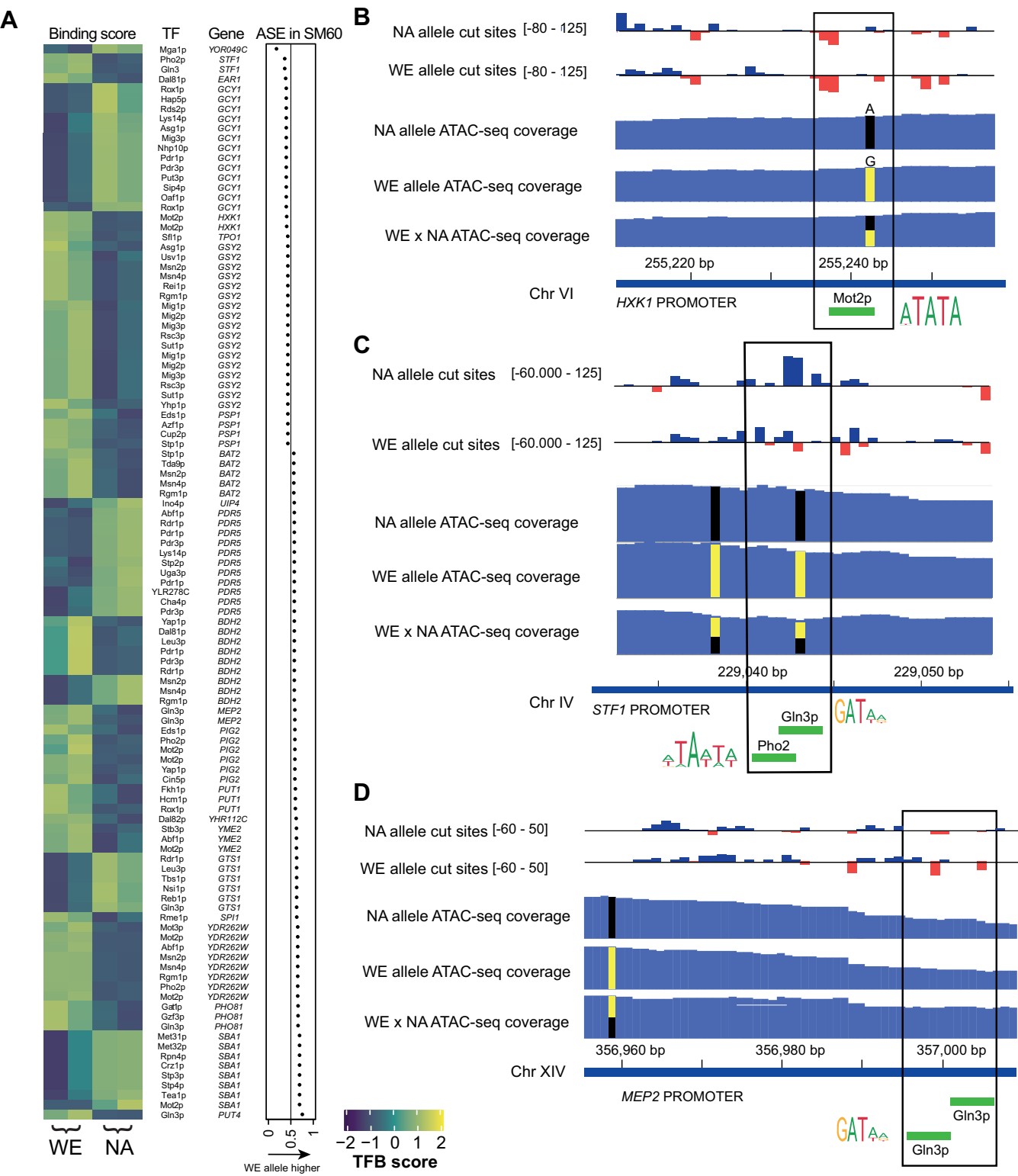

**FIG 6** ASB in genes induced under low nitrogen and showing ASE. (A) Heatmap depicting the allele-specific binding scores of 61 TFs bound to 23 regulatory regions of genes induced by low nitrogen. These genes showed allelic imbalance in gene expression (dot plot) but no allelic differences in accessibility. TFB scores are shown as z-score. (B to D) Three examples of allelic imbalance in TFB explaining ASE. The bar plot shows the bias-corrected Tn5 cut site signal obtained as reported by TOBIAS. Negative values (red bars) surrounded by positive values (blue bars) suggest the presence of a transcription factor binding footprint. The range of values for each bar plot is indicated. ATAC-seq coverage demonstrates similar accessibility in both parental backgrounds. Allele-specific data (cut sites and coverage) were obtained after splitting SNP-informative reads from the hybrid ATAC-seq alignments. Green bars show the location of predicted motifs found at differentially bound regions. Logos show the consensus motifs.

gene expression, which could still have occurred at an earlier or later time point during fermentation. Several hexose transporters are downregulated under nitrogen starvation during wine fermentation (50), with the exception of the hexose transporter *HXT5* (51), for which we also report increased accessibility and expression under low nitrogen. In addition, we found that several ribosome biogenesis genes were expressed less under low nitrogen, but this response was absent when we inspected accessibility at their promoters, suggesting that this group of genes is regulated by mechanisms other than chromatin organization. In fact, the gene expression output that we measured (mainly processed mRNA) can be influenced by mechanisms other than chromatin accessibility, including transcription rate (52), mRNA turnover (53), and by regulatory elements such as TFs and noncoding RNAs (33).

Nitrogen availability is essential for a complete wine fermentation, and nitrogen scarcity affects yeast biomass (54), fermentation performance, and time to complete fermentation (3, 55). The domestication process selected wine yeasts to withstand nitrogen scarcity stress while still maintaining good fermentation performance (14). For instance, ammonium is an excellent nitrogen source for yeast growth and is rapidly consumed at early stages, but only if the concentration of other preferred nitrogen sources such as glutamine is low (42). Here, we found that the ammonium permease *MEP2* was highly induced under low nitrogen while also showing significant allelic bias favoring the WE allele. Our findings indicate that differential binding of Gln3p (or other GATA-like TFs) at the *MEP2* promoter might explain differences in allelic expression in the absence of differential chromatin accessibility, in agreement with Gln3p being a crucial regulator of *MEP2* expression (56). A similar case was found for the nonpreferred nitrogen source proline, in which the transporter encoded by the *PUT4* gene was found in allelic bias favoring the WE allele, again in the absence of differences in chromatin accessibility. This finding is interesting, since proline cannot be assimilated under oxygen-deprived conditions (57). Indeed, oxygenation has a significant effect upon wine fermentation, accelerating the fermentation rate and impacting the production of volatile compounds (58). Additionally, we found that the allantoate permease *DAL5* has a strong allelic bias favoring the NA strain. Allantoate is absent in wine must (59), and our results point to *cis* regulation orchestrating low priority uptake of allantoate in the WE background. Summarizing, the expression of genes involved in nitrogen transport was frequently found in allelic bias, in particular when cells suffered nitrogen scarcity. The expression of these transporters is mainly controlled by two pathways, the NCR and the SPS (Ssy1-Ptr3-Ssy5) sensor system, which are differentially activated depending on nitrogen availability. In wine fermentation, the SPS pathway maximizes the uptake of preferred nitrogen sources by inducing the expression of their specific permeases (42). On the other hand, the NCR pathway represses generic permeases and those involved in internalizing poor nitrogen sources, but under nitrogen insufficiency such repression is released by NCR deactivation (42). To highlight the involvement of these two pathways in the allelic differences in expression and regulation found among genes encoding nitrogen-compound permeases, we show a summary of our findings in Fig. 7.

As highlighted herein, our results indicate an important participation of GATA TFs in *cis*-regulatory divergence driving physiological differences under low nitrogen fermentation. Regions containing GATA motifs were more likely to have higher allele accessibility and/or allelic binding, an observation that correlates with higher allelic expression. Importantly, this link between imbalance in allelic expression and regulation was significant only under nitrogen stress, highlighting the role of environmental fluctuations on *cis*-acting causal variants driving eQTLs. In addition, we identified several binding motifs affected by variants that might constitute causal polymorphisms driving differential allelic expression. Certainly, allele-specific binding data obtained from digital TF footprinting serves to identify chromatin regions likely containing causal variants driving differences in allelic expression. Here, we compared two genetic backgrounds in a hybrid context under controlled laboratory conditions; hence,

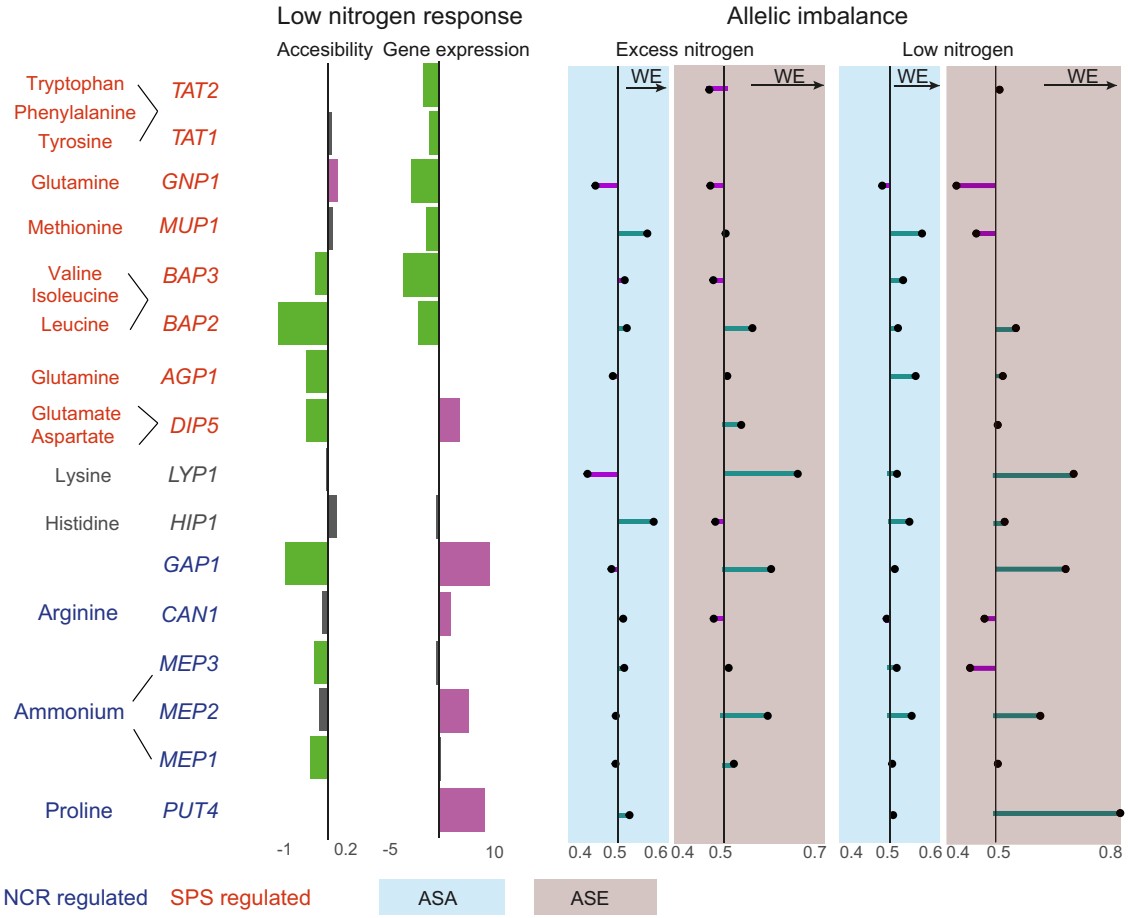

**FIG 7** Allelic imbalance profile of genes encoding permeases involved in nitrogen uptake. Summary of the allelic bias in expression and accessibility found for genes encoding transporters involved in nitrogen uptake. Genes are colored according to the pathway by which they are mainly regulated (NCR or SPS). The preferred substrates indicated for each transporter are based on those described previously (42). Data obtained from ATAC-seq or RNA-seq are shown with blue or red blackgrounds, respectively. The bars show the average fold change induction (purple) or repression (green) in response to low nitrogen. The line plots show the allelic bias favoring either the WE (> 0.5) or NA (< 0.5) allele.

mechanistic conclusions should be taken cautiously. These candidate regions that contain *cis*-regulatory variants could be subjected to future experimental validation, such as precise genome editing or allele swap, to validate their role in determining phenotypic differences.

In conclusion, we demonstrate that joint allele-specific profiling of chromatin accessibility and gene expression of a divergent yeast cross unveil regulatory dynamics driving a considerable portion of transcriptome divergence. Our findings determine the contribution of chromatin organization toward allelic differences in expression. Importantly, we detected that allele-specific TF binding adds a layer of regulation in the absence of differences in promoter accessibility. Our results improve our understanding of *cis*-regulatory elements' role on nitrogen regulation and starvation adaptation in winemaking yeast.

## MATERIALS AND METHODS

**Yeast strains and culture conditions.** We used *S. cerevisiae* strains DBVPG6765 (hereinafter referred to as Wine European [WE]) and YPS128 (hereinafter referred to as North American [NA]), as previously described (60, 61). An F1 hybrid (WE × NA) was constructed by mating haploid strains of opposite mating types of WE (*MATα hoΔ::NatMX ura3Δ::KanMX*) and NA (*MATa hoΔ::HphMX ura3Δ::KanMX*), which had been generated previously (61).

Fermentations were performed in synthetic wine must (SM) following the recipe in reference 10 with modifications (see Table S6 in the supplemental material). Two SMs containing different yeast assimilable

nitrogen (YAN) concentrations were used for fermentations, i.e., SM300 (300 mg/ml YAN [excess nitrogen]) and SM60 (60 mg/ml YAN [low nitrogen]).

**Fermentation assays.** Precultures were started from single colonies collected from YPD plates. These cultures were grown for 48 h in 50-ml Falcon tubes containing 10 ml of SM60 or SM300 at 28°C under constant agitation (250 rpm). From these precultures, $1 \times 10^6$ cells/ml were inoculated into 50 ml of SM60 or SM300 for fermentation. We conducted fermentations in at least three biological replicates in 250-ml bottles sealed with a drilled rubber stopper coupled to an airlock filled with 80% glycerol to allow for $CO_2$ release. A 100-mm cannula was inserted into the rubber stopper to perform periodical sampling. All components were autoclaved, and fermentors were assembled under sterility inside a flow cabin. Fermentations were performed under constant agitation using magnetic stirring at 25°C. Must samples were collected by obtaining 0.5 to 1 ml of fermented SM. Fermentation kinetics were monitored by manually weighing the fermentors to determine $CO_2$ release every 2 days.

**High performance liquid chromatography.** Samples obtained from fermentations were processed for quantification of amino acids using high performance liquid chromatography (HPLC) (Shimadzu, USA) with a Bio-Rad HPX-87H column (62). Briefly, 100 $\mu$l of filtered must was incubated with 3 $\mu$l of diethyl ethoxymethylenemalonate (DEEM) (Sigma-Aldrich catalog no. 87-13-8) for 30 min in a sonication bath at room temperature in a solution containing 580 $\mu$l of borate buffer (pH 9) and 250 $\mu$l of methanol. After sonication, samples were incubated for 2 h at 70°C. After the derivatization reaction, 20 $\mu$l of the processed samples was injected into a Shimadzu Prominence HPLC (Shimadzu, USA). The concentration of each amino acid was calculated using a calibration curve obtained from sequential dilutions of nonfermented SM.

**RNA extraction and sequencing.** Yeast cells of the WE $\times$ NA hybrid were collected for RNA sequencing after 14 h of fermentation in SM300 or SM60 in triplicates. Yeast cells were washed three times with phosphate-buffered saline (PBS) buffer, and total RNA was extracted following a hot-formamide protocol (63). RNA in formamide was treated with DNase I (Promega, USA) to remove genomic DNA traces and purified using the GeneJET RNA Cleanup and Concentration Micro kit (Thermo Fisher Scientific catalog no. K0841). RNA integrity was confirmed using a Fragment Analyzer (Agilent, USA). The RNA-seq libraries were constructed using the TruSeq RNA Sample Prep kit v2 (Illumina, USA). Sequencing was conducted using paired-end 100-bp reads on an Illumina HiSeq in a single lane for all samples.

**ATAC-seq assay and sequencing.** For ATAC-seq, fermentations of the WE and NA parental strains, together with the WE $\times$ NA hybrid, were performed in duplicates, and cells in SM60 and SM300 were sampled after 14 h. Cells were quantified using a Neubauer counting chamber. Five million haploid WE and NA cells and 2.5 million diploid WE $\times$ NA cells were spun down (1.8 $\times$ g for 4 min at room temperature) and washed twice using SB buffer (1 M sorbitol, 10 mM $MgCl_2$, 40 mM HEPES [pH 7.5]). Cells were treated with 50 mg/ml of zymolyase 20T (Euromedex UZ1000-A) in 200 $\mu$l of SB for 30 min at 30°C, after which cells were washed two times with SB buffer. Immediately after, cells were incubated for 30 min at 37°C in 50 $\mu$l of transposition mix, containing 25 $\mu$l Nextera Tagment DNA Buffer (Illumina catalog no. 15027866), 22.5 $\mu$L $H_2O$, and 2.5 $\mu$l Nextera Tagment DNA enzyme I (Illumina catalog no. 15027865). Subsequently, DNA was purified using the DNA Clean and Concentrator-5 kit (Zymo Research D4003) following the supplier's instructions.

Tagmented DNA was amplified and barcoded using Nextera Index i5 and i7 series PCR primers. The PCR consisted of 25 $\mu$l NEBNext Hi-Fidelity 2$\times$ PCR Master Mix (New England Biolabs catalog no. NEB. M0541S), 7.5 $\mu$l $H_2O$, 6.25 $\mu$l i5 primer (10 mM), 6.25 $\mu$l i7 primer (10 mM), and 5 $\mu$l tagmented DNA. PCR cycles were set as follows: 1 cycle of 72°C for 5 min; 1 cycle of 98°C for 30 s; 8 cycles with 1 cycle consisting of 98°C for 10 s, 63°C for 30 s, and 72°C for 1 min. Subsequently, the amplified ATAC-seq library was subjected to double-sided size selection using magnetic beads (AMPure XP; Beckman Coulter catalog no. BC-A63880). First, 50 $\mu$l of the library was incubated with 20 $\mu$l of beads (0.4$\times$), after which the supernatant was collected. Subsequently, a left-side selection was performed by incubating the library with 1.1$\times$ of beads, after which the supernatant was discarded. DNA bound to the beads was washed twice with freshly made 80% ethanol and then eluted in 20 $\mu$l of $H_2O$. Library quality was assessed using a Fragment Analyzer (Agilent, USA) and quantified in Qubit (Thermofisher, USA). Sequencing was conducted using paired-end 75-bp reads on an Illumina NextSeq 500.

**Allele-specific read mapping.** To estimate allele-specific counts derived from RNA-seq and ATAC-seq reads and to account for the mapping of SNP-informative reads, we modified the *S. cerevisiae* reference genome (R64-1-1) using SNPsplit (64). Genome-wide SNP data from the WE and NA strains were obtained from the *Saccharomyces* Genome Resequencing Project (65). These data were used to replace the reference genome nucleotide sequence at 17,425 sites, in which the same genotype occurs for these two strains, but differed against the S288c reference strain. Next, the modified reference sequence was masked at 81,169 polymorphic sites between the NA and WE strains. A genome index was built using Bowtie2 (66) and then used to map ATAC-seq and RNA-seq reads in the WE $\times$ NA hybrid (using Bowtie2 option "very-sensitive"). Before allele-specific mapping, sequencing reads were processed using fastp (67) to trim low quality 3' ends (Q < 20) and to exclude reads shorter than 36 bp (-A, disable automatic adaptor trimming; -3, trimming by quality at 3' end; -l 36, min length of 36 bp). Alignments containing all hybrid mapped reads were used for differential analysis of gene expression and chromatin accessibility of the WE $\times$ NA hybrid response to low nitrogen. Furthermore, SNPsplit was used to divide the hybrid alignments in two bam files, each containing only SNP-informative reads corresponding to either parental background (approximately 16 to 22% of the total mapped reads for each parent [Table S7]). These allele-specific alignment files were used to evaluate the response (gene expression, chromatin accessibility, and TF binding) of each parental genome to low nitrogen.

**Hybrid RNA-seq analysis.** RNA-seq read counts per gene in the WE × NA hybrid were obtained from bam alignments using featureCounts (68) and the modified R64-1-1 genome annotation (ENSEMBL, release date 2018-10). Genes with at least 30 read counts across the three replicates under at least one condition were selected for further statistical analysis (5,625 out of 6,534 genes). The differential transcriptome response of the WE × NA hybrid to low nitrogen (SM60) was estimated using DESeq2 (69) (design= ~ condition).

**ATAC-seq data analysis.** ATAC-seq read alignments were processed as follows. (i) PCR duplicates were identified and removed using Genrich (github.com/jsh58/Genrich). (ii) Reads mapped to blacklisted regions were removed (i.e., mitochondrial genome, ribosomal genes in chromosome 12, and subtelomeres). (iii) Only properly paired mapped reads (mates mapped to the same chromosome, pairs mapped in convergent direction) were kept using sambamba (70). ATAC-seq coverages around the transcription start site (TSS) of those genes that passed RNA-seq count filters were obtained using deepTools computeMatrix (10-kb bins) and plotHeatmap (71), which were then further processed for plotting with the ComplexHeatmap R package (72). TSSs were obtained from reference 73. Correlations between ATAC-seq coverage (RPKM) and gene expression (RPKM) were calculated using the cor R package (method = "spearman"). To match genes with their nearby ATAC-seq signal, we selected a regulatory region 400 bp upstream of the TSS for each gene. For 224 genes that lacked an annotated TSS, we used the transcript start as TSS. The ATAC-seq signals of 5,625 regulatory regions were quantified by counting mapped reads using featureCounts. Differential responses in ATAC-seq in the WE × NA hybrid to low nitrogen (SM60) were estimated using DESeq2 (design= ~ condition).

**Allelic imbalance analyses of RNA-seq and ATAC-seq data.** To test for differential allelic imbalance in chromatin accessibility or gene expression, we used the R package MBASED on allele counts (74). Counts at each SNP were obtained using ASEReadCounter employing the pipeline described in reference 77. We excluded those SNPs from one parent that overlapped with an indel of the other parental strain and those that fell within low mappability regions of either parent (determined after mapping parental DNA-seq reads to the reference genome). Besides, only SNPs with at least five counts in both parents in SM300 or SM60 were retained for further analysis. Allelic counts were obtained by summing up SNP read counts at regulatory regions (ATAC-seq) or genes (RNA-seq). A binomial test implemented in MBASED was used to statistically evaluate allelic imbalance. Genes or regulatory regions with an adjusted $P$ value (Benjamini-Hoch [BH] correction) lower than 0.05 in at least two replicates were considered to display significant allelic imbalance.

**Allele-specific transcription factor binding analyses from ATAC-seq footprints.** Analysis of allele-specific transcription factor binding (TFB) from ATAC-seq footprints was performed using TOBIAS (78). Briefly, ATAC-seq alignments of the WE × NA hybrid were used to obtain Tn5 cut sites which were corrected for cutting bias by TOBIAS ATACorrect. The binding signal per site was calculated using TOBIAS ScoreBigWig (–fp-min 5 –fp-max 30) for 5,401 regulatory regions. To calculate TFB, we obtained binding motifs for 141 yeast TFs from the JASPAR database (75). Based on this, TFB was quantified at motifs occurring in the regulatory regions by TOBIAS BinDetect. Binding scores were further processed in R. To evaluate differences in binding scores, we used a linear model implemented by the limma R package (76). We excluded from this analysis those TFs that showed low expression in SM300 or SM60 (RPKM < 5), and motif sites that were not considered by TOBIAS as "bound" in any condition. We considered binding differences with a FDR < 0.1 as statistically significant. For the analysis of allele specific TFB, allele-specific mapped reads were identified with SNPsplit (see above). Allele-specific alignments were further processed for TOBIAS analyses, as previously indicated for the hybrid. In allele-specific analysis, we excluded motifs located at regions having low ATAC-seq coverage after SNP splitting. Total and allele-specific alignment statistics are provided in Table S7.

**Data availability.** RNA-seq and ATAC-seq raw reads are available in SRA under the project PRJNA705961.

## SUPPLEMENTAL MATERIAL

Supplemental material is available online only.
**FIG S1**, PDF file, 0.1 MB.
**FIG S2**, PDF file, 0.9 MB.
**FIG S3**, PDF file, 0.9 MB.
**TABLE S1**, XLSX file, 0.3 MB.
**TABLE S2**, XLSX file, 0.02 MB.
**TABLE S3**, XLSX file, 1.2 MB.
**TABLE S4**, XLSX file, 0.02 MB.
**TABLE S5**, XLSX file, 0.4 MB.
**TABLE S6**, XLSX file, 0.01 MB.
**TABLE S7**, XLSX file, 0.01 MB.

## ACKNOWLEDGMENTS

We are grateful to Valentina Abarca and Antonio Molina for their assistance in maintaining yeast collections. We thank J. M. Alvarez for his assistance with chromatin

accessibility analyses and C. Meneses (UNAB) for his support in ATAC library construction and sequencing. We thank Michael Hanford (Universidad de Chile) for language support.

F.A.C. was supported by Comisión Nacional de Investigación Científica y Tecnológica CONICYT FONDECYT (1180161) and ANID - Programa Iniciativa Científica Milenio - ICN17_022. C.A.V. was supported by CONICYT FONDECYT (3170404). The funders had no role in study design, data collection and analysis, decision to publish, or preparation of the manuscript.

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
