## [Reviewer comments · mSystems]

Uncovering divergence in gene expression regulation in the adaptation of yeast to nitrogen scarcity

Carlos Villarroel, Paulo Canessa, Macarena Bastias, and Francisco Cubillos

Corresponding Author(s): Francisco Cubillos, Universidad de Santiago de Chile

Review Timeline:

Submission Date:	April 14, 2021
Editorial Decision:	July 1, 2021
Revision Received:	July 22, 2021
Accepted:	August 5, 2021

Editor: Jeffrey Blanchard

Reviewer(s): The reviewers have opted to remain anonymous.

Transaction Report:

DOI: <https://doi.org/10.1128/mSystems.00466-21>

July 1, 2021

Dr. Francisco A Cubillos
Universidad de Santiago de Chile
Santiago
Chile

Re: mSystems00466-21 (Uncovering divergence in gene expression regulation in the adaptation of yeast to nitrogen scarcity)

Dear Dr. Francisco A Cubillos:

Thank you for submitting your manuscript to mSystems. We have completed our review and I am pleased to inform you that, in principle, we expect to accept it for publication in mSystems. However, acceptance will not be final until you have adequately addressed the reviewer comments.

After reading the manuscript myself, I have decided to move forward with just 2 reviews instead of waiting longer for the third reviewer (reviewer #1). There are several general considerations raised by each reviewer. Please give these some thought in your response. Although I personally am comfortable with the techniques and bioinformatic analyses, the writing is very technical and you may have trouble engaging scientists interested in the topic.

Preparing Revision Guidelines

For complete guidelines on revision requirements for your article type, please see the journal Article Types requirement at <https://journals.asm.org/journal/mSystems/article-types>. **Submissions of a paper that does not conform to mSystems guidelines will delay acceptance of your manuscript.**

Corresponding authors may join or renew ASM membership to obtain discounts on publication fees.

Need to upgrade your membership level? Please contact Customer Service at Service@asmusa.org.

Sincerely,

Jeffrey Blanchard

Editor, mSystems

Journals Department
Reviewer comments:

Reviewer #2 (Comments for the Author):

To survive to stressful conditions such as nitrogen scarcity, the budding yeast *Saccharomyces cerevisiae* rewires its transcriptional output. In this study, the authors used a hybrid generated with two isolates (a winemaking and oak tree strain) showing differences in nitrogen consumption in order to map cis factors involved in the divergence of gene expression in response to nitrogen scarcity. In this context, they obtained genome-wide allele-specific profiles of chromatin accessibility, transcription factor binding, and gene expression through ATAC seq and RNA-seq. They evaluated whether differences in nitrogen consumption between the strains were due to cis-regulatory variants modulated through environments differing in nitrogen availability. They reported numerous events of allelic differences in chromatin accessibility between these two strains, with few of them correlating with ASE. In fact, one third of the allelic differences in gene expression and accessibility only occur under low nitrogen. By performing allele specific TFB footprinting, they revealed potential TFs driving allelic expression differences, some of which have never been associated to the regulation of nitrogen metabolism.

The results presented in this manuscript are clear and the conclusions are convincing and well supported by facts. The experiments are carefully designed and well described so they can be reproduced. In particular, experiments are thoroughly carried out and provide very detailed information. Finally, the results are clearly discussed and compared to previous studies related to the topic. Overall, the paper is well-written and the figures and tables clear and informative. It is within the scope of mSystems.

Nevertheless, I have several points that need to be addressed:

1. The authors crossed two isolates, a winemaking strain (DBVPG6765), and an un-domesticated

strain (YPS128) isolated from an oak tree. They said that these strains are divergent and contrasting.

How divergent are these strains in terms of genetic divergence? In addition, variation in nitrogen metabolism has been described among large populations of isolates. It would be interesting to replace this study and compare the phenotypic behavior of these two strains with the others.

2. page 5 line 129

The authors say 'we used WE xNA...' but they never defined WE and NA in the main text. Which strain is which?

3. page 5 line 144

The authors mentioned YAN but as previously this is not defined before in the main text. Only in the figure legend.

4. page 6 line 160

"We collected mRNA from the WE x NA hybrid after 14 hours of fermentation"
Why after 14 hours?

5. page 7 line 190

"we considered a regulatory region of 400 bp upstream of the TSS of each gene"
Here also, why 400 bp?

6. In the discussion, the authors should mention the fact that the study was based on one hybrid generated with two genetic backgrounds and consequently some conclusions should be taken with caution.

Reviewer #3 (Comments for the Author):

The manuscript by Villarroel et al. contains experiments that have been carefully performed and the conclusions supported by the data presented. There are, however, several points that would benefit from attention.

1. The manuscript is largely descriptive in nature. The authors have presented evidence on a genome wide scale that has been repeatedly shown to occur on a gene specific basis.

2. The manuscript's greatest value is not in its conclusions but as a source of data to be mined by others.

3. The manuscript will be difficult to read for investigators outside of the systems biology field thereby reducing its overall impact.

4. In multiple places the authors cite gene names without identifying their designations, e.g., line 369. When a gene name is used, it would be useful to the reader to also identify its function.

5. It has been shown that DAL80 binds to multiple loci within the coding region of many genes. It is somewhat surprising that these interactions were not analyzed.

6. It may be beyond the scope of the present work, but none of the conclusions of the work were validated in an independent manner using newly discovered genes showing the greatest effects in their experiments and designated as being new elements in nitrogen-responsive regulation.

Please see comments to the authors for information pertinent to questions 1 and 2.

Response to Reviewers

After reading the manuscript myself, I have decided to move forward with just 2 reviews instead of waiting longer for the third reviewer (reviewer #1). There are several general considerations raised by each reviewer. Please give these some thought in your response. Although I personally am comfortable with the techniques and bioinformatic analyses, the writing is very technical and you may have trouble engaging scientists interested in the topic.

R: We thank the editor for providing the available reviews. In this new version, we reduced some technical aspects throughout the manuscript and we aimed to provide more biological insights (for example in the Discussion section) into the text. We sincerely hope that this new version meets the requirements made by both reviewers and the editor.

Reviewer comments:

Reviewer #2 (Comments for the Author):

To survive to stressful conditions such as nitrogen scarcity, the budding yeast *Saccharomyces cerevisiae* rewires its transcriptional output. In this study, the authors used a hybrid generated with two isolates (a winemaking and oak tree strain) showing differences in nitrogen consumption in order to map cis factors involved in the divergence of gene expression in response to nitrogen scarcity. In this context, they obtained genome-wide allele-specific profiles of chromatin accessibility, transcription factor binding, and gene expression through ATAC seq and RNA-seq. They evaluated whether differences in nitrogen consumption between the strains were due to cis-regulatory variants modulated through environments differing in nitrogen availability. They reported numerous events of allelic differences in chromatin accessibility between these two strains, with few of them correlating with ASE. In fact, one third of the allelic differences in gene expression and accessibility only occur under low nitrogen. By performing allele specific TFB footprinting, they revealed potential TFs driving allelic expression differences, some of which have never been associated to the regulation of nitrogen metabolism.

The results presented in this manuscript are clear and the conclusions are convincing and well supported by facts. The experiments are carefully designed and well described so they can be reproduced. In particular, experiments are thoroughly carried out and provide very detailed information. Finally, the results are clearly discussed and compared to previous studies related to the topic. Overall, the paper is well-written and the figures and tables clear and informative. It is within the scope of mSystems.

Nevertheless, I have several points that need to be addressed:

1. The authors crossed two isolates, a winemaking strain (DBVPG6765), and an un-domesticated strain (YPS128) isolated from an oak tree. They said that these strains are divergent and contrasting.

How divergent are these strains in terms of genetic divergence?

R: The two strains used in this study exhibit an average of 1 SNP every 148 bp, equivalent to 0.6% sequence divergence. In total, there are 76,727 SNPs between WE and NA, which

represents a greater number of SNPs when compared to the widely used BY × RM cross, which has ~47,000 of segregating sites. (Natural single-nucleosome epi-polymorphisms in yeast. PLoS Genet. 2010 Apr 22; 6(4):e1000913.)

We have now included the sequence divergence percentage in the main text (between current lines 129 and 138).

In addition, variation in nitrogen metabolism has been described among large populations of isolates. It would be interesting to replace this study and compare the phenotypic behavior of these two strains with the others.

R: Indeed, these two strains have contrasting nitrogen consumption profiles. Others (and ourselves) previously studied these strains in synthetic wine musts, demonstrating that wine strains have efficient consumption profiles compared to oak isolates. These previous studies led us to select these two strains to construct the F1 hybrid and evaluate allele-specific differences for this phenotype. Yet, this is the first study to compare these wine and oak strains consumption profiles under low nitrogen conditions. We have now referred to those manuscripts in several passages of main text.

2. page 5 line 129

The authors say 'we used WE xNA...' but they never defined WE and NA in the main text. Which strain is which?

R: This is now indicated in the main text (now lines 162-163)

3. page 5 line 144

The authors mentioned YAN but as previously this is not defined before in the main text. Only in the figure legend.

R: This is now defined in the main text (now line 168)

4. page 6 line 160

"We collected mRNA from the WE x NA hybrid after 14 hours of fermentation"
Why after 14 hours?

R: We thank the reviewer for this question. We have now included a new sentence in the manuscript to clarify explain why the 14 hours time-point was chosen (now lines 211-215)

'We chose this time point due to the primary consumption of preferred nitrogen sources under nitrogen excess (NCR suppressed state) and the complete YAN consumption under low nitrogen, likely triggering a nitrogen starvation stress response (NCR active state). Hence significant differences in gene expression and regulation between environments and genetic backgrounds were expected'

5. page 7 line 190

"we considered a regulatory region of 400 bp upstream of the TSS of each gene"
Here also, why 400 bp?

R: Again, we appreciate the reviewer's observation. This is now explained in the main text. We also refer to another study where the same criteria for regulatory regions was used to evaluate chromatin accessibility. (please see new text between lines 240-243).

'For the analyses shown hereafter, to evaluate the corresponding ATAC-seq signal, we considered a regulatory region of 400 bp upstream of the TSS of each gene. We based the selection of the regulatory region size on two antecedents, i) it correlates well with gene expression in low and excess nitrogen conditions, and ii) another study previously used a similar region size'

6. In the discussion, the authors should mention the fact that the study was based on one hybrid generated with two genetic backgrounds and consequently some conclusions should be taken with caution.

We strongly agree with the reviewer, in particular when discussing mechanistic events driving ASE such as GATA TFs that might be specific for these two genetic backgrounds. We have now incorporated the following sentence to address this claim.

'Here, we compared two genetic backgrounds in a hybrid context under controlled laboratory conditions; hence mechanistic conclusions should be taken cautiously'

Reviewer #3 (Comments for the Author):

The manuscript by Villarroel et al. contains experiments that have been carefully performed and the conclusions supported by the data presented. There are, however, several points that would benefit from attention.

1. The manuscript is largely descriptive in nature. The authors have presented evidence on a genome wide scale that has been repeatedly shown to occur on a gene specific basis.

R: This manuscript intends to provide novel insights into mechanisms underlying nitrogen consumption differences between yeast strains. In this way, we identified biological and molecular processes differentially expressed between strains, providing an alternative perspective to understand phenotypic variability due to mutations in regulatory regions (rather than only mutations in coding regions). Furthermore, we provide specific examples of genes and TFs to give the reader a closer perspective of what happens at the gene level. Yet, novel cis-mechanisms driving the adaptation to low nitrogen fermentation are provided, demonstrating an environmental-specific cis-response.

2. The manuscript's greatest value is not in its conclusions but as a source of data to be mined by others.

R: We appreciate that others could consistently use this dataset in the future, and we expect to serve as a contribution to the scientific community trying to assess cis-regulatory variation.

3. The manuscript will be difficult to read for investigators outside of the systems biology field thereby reducing its overall impact.

We still expect that our results can evoke interest beyond the system biology field, although, we think that they are very suitable for the mSystems readership. For example, some of our findings complements previous research in genetic variants affecting wine fermentation, which currently are constrained to coding genes. In addition, ATAC-seq has become a technology heavily used in mammal research, but not widely exploited in eukaryotic microorganisms, hence we expect that our research can guide future studies in other models.

4. In multiple places the authors cite gene names without identifying their designations, e.g., line 369. When a gene name is used, it would be useful to the reader to also identify its function.

R: We thank the reviewer for this suggestion. We have now expanded gene information regarding their designations for THI4, STF1 and others (e.g., please see new lines 452 and 456). Proteins mentioned only by name corresponded to transcription factors.

5. It has been shown that DAL80 binds to multiple loci within the coding region of many genes. It is somewhat surprising that these interactions were not analyzed.

R: We agree with the reviewer that the findings in Ronsmans et al (2019), i.e., DAL80 binding at coding regions of highly expressed NCR-responsive genes, are interesting events occurring during nitrogen stress in yeast. However, those events are rare, and accounting for them could further complicate our genome-wide testing of ATAC-seq footprints. Importantly, we have cross-referenced our data with those of Ronsmans et al (2019) (please, see new lines 318-321). Following up on the reviewer's suggestion, we tested for footprints at the MEP2 coding region (studied in detail in Ronsmans et al.). However, we did not find any significant scores for any footprint, probably due to low ATAC-seq coverage at the MEP2 coding region.

6. It may be beyond the scope of the present work, but none of the conclusions of the work were validated in an independent manner using newly discovered genes showing the greatest effects in their experiments and designated as being new elements in nitrogen-responsive regulation.

R: We agree with the reviewer that wet-lab experiments could have further supported and/or expanded some of our findings. In fact, in our manuscript, we acknowledge this minor limitation (e.g., lines 450-453), which, as the reviewer points out, does not affect the current scope of our work, but instead encourages different and exciting paths for the future. In addition, due to last year limitations to lab work, we decided to address mechanistic questions using additional bioinformatic analyses beyond those usually performed in ATAC-seq studies, such as allele-specific TF footprints analysis.

August 5, 2021

Dr. Francisco A Cubillos
Universidad de Santiago de Chile
Santiago
Chile

Re: mSystems00466-21R1 (Uncovering divergence in gene expression regulation in the adaptation of yeast to nitrogen scarcity)

Dear Dr. Francisco A Cubillos:

Thank you for your thoughtful comments. Your manuscript has been accepted, and I am forwarding it to the ASM Journals Department for publication. For your reference, ASM Journals' address is given below. Before it can be scheduled for publication, your manuscript will be checked by the mSystems senior production editor, Ellie Ghatineh, to make sure that all elements meet the technical requirements for publication. She will contact you if anything needs to be revised before copyediting and production can begin. Otherwise, you will be notified when your proofs are ready to be viewed.

As an open-access publication, mSystems receives no financial support from paid subscriptions and depends on authors' prompt payment of publication fees as soon as their articles are accepted. =

Publication Fees:

We recognize that the video files can become quite large, and so to avoid quality loss ASM

suggests sending the video file via <https://www.wetransfer.com/>. When you have a final version of the video and the still ready to share, please send it to Ellie Ghatineh at eghatineh@asmusa.org.

Sincerely,

Jeffrey Blanchard
Editor, mSystems

Journals Department
Table S4: Accept
Figure S1: Accept
Table S2: Accept
Figure S2: Accept
Table S6: Accept
Table S5: Accept
Table S3: Accept
Figure S3: Accept
Table S1: Accept
Table S7: Accept